# Phytosterols: Physiological Functions and Potential Application

**DOI:** 10.3390/foods13111754

**Published:** 2024-06-03

**Authors:** Mingyue Shen, Lanlan Yuan, Jian Zhang, Xufeng Wang, Mingyi Zhang, Haizhen Li, Ying Jing, Fengjiao Zeng, Jianhua Xie

**Affiliations:** State Key Laboratory of Food Science and Resources, Nanchang University, Nanchang 330047, China; shenmingyue1107@ncu.edu.cn (M.S.); 15270874205@163.com (L.Y.); zj105716@163.com (J.Z.); 357900230032@email.ncu.edu.cn (X.W.); 407900220048@email.ncu.edu.cn (M.Z.); lhz.19496@email.ncu.edu.cn (H.L.); ncujingying@email.ncu.edu.cn (Y.J.); zfj2405yeah@163.com (F.Z.)

**Keywords:** phytosterols, function, cholesterol-lowering, anti-inflammatory, application

## Abstract

Dietary intake of natural substances to regulate physiological functions is currently regarded as a potential way of promoting health. As one of the recommended dietary ingredients, phytosterols that are natural bioactive compounds distributed in plants have received increasing attention for their health effects. Phytosterols have attracted great attention from scientists because of many physiological functions, for example, cholesterol-lowering, anticancer, anti-inflammatory, and immunomodulatory effects. In addition, the physiological functions of phytosterols, the purification, structure analysis, synthesis, and food application of phytosterols have been widely studied. Nowadays, many bioactivities of phytosterols have been assessed in vivo and in vitro. However, the mechanisms of their pharmacological activities are not yet fully understood, and in-depth investigation of the relationship between structure and function is crucial. Therefore, a contemporaneous overview of the extraction, beneficial properties, and the mechanisms, as well as the current states of phytosterol application, in the food field of phytosterols is provided in this review.

## 1. Introduction

Phytosterols are natural plant-derived compounds widely distributed in plants with similar chemical structure and physiological functions to cholesterol [1], but they differ in side chains and ring structure saturation [2]. Phytosterols can be divided into free form or they can be covalently bound via an ester or glycosidic bond [3]. The differences in structures of phytosterols are well related to their bioactivities [4]; it is known that the most common locations of functional groups are on C_3_, C_4_, C_7_, C_12_, and C_17_ (Figure 1) [5].

In recent years, phytosterols have drawn a lot of attention due to their outstanding physiological activity. Phytosterols possess many physiological activities such as cholesterol-lowering, anti-inflammatory, and anticancer effects. Long-term consumption of phytosterol-rich foods can have a 20% reduction in the risk of cancer [6]. In addition, phytosterols have been found to be great preventives of cardiovascular and cerebrovascular diseases in humans [7]. Humans cannot synthesize phytosterols on their own and mainly consume them from vegetable oils, nuts, and mushrooms [8]. A total sterol intake of about 150 to 400 mg/day has been reported in our daily diet, consisting mainly of β-sitosterol, campesterol, and soya sterols [7]. The absorption rate of phytosterols in the intestine is very low in the human body [9]; levels of phytosterols in serum of humans range from 2.9 to 17.0 mg/L [10]. A moderate increase in daily phytosterols intake is found to be beneficial to human health; it is indicated that intake of 2 g/day of phytosterols will reduce levels of low-density lipoprotein cholesterol, ranging from 7 to 10% [11].

This review focuses on the recent advances in the extraction process of phytosterols, the physiological activities of phytosterols, including cholesterol-lowering, anticancer, anti-inflammatory, antioxidative, and immunomodulatory activities, and the potential food application prospects of phytosterols in functional foods are also prospected. These will be helpful for the developments of functional foods of phytosterols.

## 2. Extraction of Phytosterols

### 2.1. Sources of Phytosterols and Extraction Techniques

Phytosterols are present in all plant cell membranes and are especially enriched in vegetable oils and fats, cereals and cereal products, vegetables, fruits, and berries [3]. In industry, phytosterols can be recovered from vegetable oil through a refining process called deodistillation [12]. Franks et al. [13] extracted 4 phytosterols from 14 spent coffee grounds from 12 different countries, which are β-sitosterol, campesterol, stigmasterol, and cycloartenol, in the order of abundance.

The isolation techniques for phytosterols rely on the nature of the matrix and the form of phytosterols (free, esterified, and glycosylated) [14]. Therefore, a significant portion of the raw material has phytosterols in an esterified form, which requires hydrolysis before extraction to release free phytosterols. Two ways of hydrolyzing are always available: high-temperature and high-pressure hydrolysis and alkali hydrolysis. The latter is more commonly used because of mild conditions, and this method allows simultaneous hydrolysis and saponification [14].There are conventional and nonconventional extraction techniques in phytosterols extraction [15]. The conventional techniques, such as soxhlet extraction, maceration [14], cold press [16], and screw pressing [17], are used for extraction of phytosterols. Among them, soxhlet extraction and cold press extraction are used commonly [18]. Due to the many limitations of conventional extraction methods, people have introduced the nonconventional techniques to overcome these limitations [16]; the nonconventional techniques include supercritical fluid extraction, enzymatic extraction, solid phase extraction, direct hydrolysis extraction, microwave-assisted hydrodistillation, silica gel adsorption, and ultrasound-assisted extraction [14,15,19].

### 2.2. Analytical Identification of Phytosterols

The entire process includes not only extraction but also saponification, refining [20], analytical identification, and purification; all parameters have to be optimized to obtain a high-quality extract with high quantities of the desired compound [14]. It is worth mentioning that the high-speed countercurrent chromatography and high-preparative-performance liquid chromatography are often used for the separation and purification of phytosterols [21]. After extraction and purification, phytosterols are usually analyzed by gas chromatography, gas chromatography–mass spectrometry, and nuclear magnetic resonance spectrum to elucidate their structure and purity [22]. Gas chromatography (GC) technology is widely used in the determination of phytosterols and becomes more accurate and efficient along with continuous development and updating [20]. High-performance liquid chromatography–mass spectrometry (HPLC–MS) is used to accurately identify and quantify phytosterols, especially to measure the molecular weight of the various phytosterol derivatives [23]. Meiko et al. [24] established a method based on HPLC with fluorescence detection for the simultaneous analysis of phytosterols (stigmasterol, β-sitosterol, campesterol, ergosterol, and fucosterol); by using this method, analysis of sterols can be achieved at a lower cost than by GC–MS and LC–MS. And liquid chromatography-atmospheric pressure chemical ionization–mass spectrometry (LC–APCI–MS) was used to identify and quantify the phytostrols from soybean, palm, sunflower, and olive [25].

## 3. Physiological Function of Phytosterols

### 3.1. Cholesterol-Lowering Effects

Cholesterol is a kind of the unsaturated alcohol in the steroid family and it is also one of the essential components of all animal cell membranes [26]. Although cholesterol is essential for the body, disturbances in cholesterol homeostasis can lead to a variety of diseases such as cardiovascular disease, diabetes, nonalcoholic fatty liver disease, cancer, and neurodegenerative diseases [27]. Therefore, purposeful control of cholesterol levels in the body is a very beneficial measure for health, and controlling cholesterol levels in a safe way without side effects has become a hot topic of research for scientists.

Cholesterol is mainly divided into exogenous and endogenous cholesterol, and exogenous cholesterol is mainly ingested through the diet, while endogenous cholesterol mainly comes from acetyl-CoA synthesis and the secretion of bile in the liver [28]. The cholesterol from bile source is in the free form, while part of dietary cholesterol is in the form of esterification, which must be transformed to free cholesterol through hydrolysis in the digestive juice. After micellar solubilization of free cholesterol in the intestinal lumen, Niemann-Pick C1-like 1 (NPC1L1) protein mediates cholesterol transport from intestinal lumen into the cells [29], and most of the absorbed cholesterol in cells is catalyzed and esterified by acyl-coenzyme A: cholesterol acyltransferase (ACAT) in the endoplasmic reticulum, forming cholesterol ester (CE) and entering the lymphatic system [28,30]. In addition, small amounts of unabsorbed cholesterol are excreted back into the intestine lumen by adenosine triphosphate binding cassette transporter G family 5/8 (ABCG 5/8) in intestinal epithelial cells (Figure 2) [31,32].

Furthermore, cholesterol cannot exist in free form in the blood, while it can only be carried by lipoproteins as the bound form. About 70% of cholesterol in the blood is carried by low-density lipoproteins (LDLs) and very low-density lipoproteins (VLDLs). The clearance of LDL cholesterol is mainly mediated by the LDL receptor (LDLR) in the liver, which is a transmembrane glycoprotein that binds to LDL, and cholesterol ester is endocytosed into the cell’s acidic endosome. Finally, LDL is degraded into amino acids, while cholesterol ester is hydrolyzed by acid esterase into free cholesterol and enters the free cholesterol pool in the cytoplasm [33]. Therefore, LDLR plays a key role in regulating the homeostasis of cholesterol.

As shown in Table 1, the cholesterol-lowering effects of phytosterols have been extensively studied, and the mechanism has also become a hot spot. Firstly, phytosterols are more hydrophobic than cholesterol; they may replace cholesterol in micelles in intestine, resulting in decreased cholesterol in micelles, reduced cholesterol absorption in intestinal lumen, and increased cholesterol excretion in feces (Figure 3a) [34]. For this mechanism, it was reported that pine-derived phytosterols supplement was effective at inhibiting cholesterol transport in co-micelle formation in an artificial intestine fluid as well as in the potential oral administration test [35]. In addition, NPC1L1 plays a key role in the absorption of cholesterol in intestine and liver, and phytosterols will affect the expression of NPC1L1 (Figure 3b). It was found that dietary addition of β-sitosterol significantly downregulated the expression mRNA of NPC1L1 in the intestine of Male Golden Syrian hamsters [36]. The NPC1L1 expression in HepG2 cells was significantly reduced after treatment with 100 μmol/L sitosterol or stigmasterol [37]. Free phytosterols can increase the expression of ABCG 5/8 in intestinal cell, which promotes the repump of cholesterol from intestinal cells into the cavity, thus reducing the absorption rate of cholesterol [38]. Brufau et al. [39] fed mice with phytosterols (69% β-sitosterol, 15.7% campesterol, and 15.7% β-sitostanol) and found that phytosterols increased cholesterol excretion by increasing the expression of ABCG5. Li et al. found that dietary supplementation of soybean sterols can be used to metabolize cholesterol by upregulating the mRNA expression levels of ABCG5, ABCG8, and NPC1L1 in intestinal tract. And the GC–MS results of fecal sterol content indicate that upregulated ABCG5/8 promotes sterol efflux [40]. Furthermore, phytosterols could interfere with the formation of chylomicron (Figure 3c): phytosterols can affect the formation of low-density lipoprotein (LDL) in blood by inhibiting the generation of Apo B in hepatocytes and reducing the content of low-density lipoprotein (VLDL). LDLR is a cell-surface glycoprotein that regulates the uptake of cholesterol-carrying lipoproteins through receptor-mediated endocytosis and plays a crucial role in maintenance of cholesterol homeostasis [41]. Phytosterols can increase the expression of LDLR, thus removing LDL-C. Chen et al. [42] fed hamsters with algal sterols and β-sitosterol and found that the expression of LDLR in liver increased significantly. On the other hand (Figure 3d), phytosterols can regulate the activities of key enzymes involved in cholesterol synthesis and metabolism, phytosterols can suppress the activity of sterol regulatory element-binding proteins 2 (SREBP 2) and cholesterol synthesis rate-limiting enzyme (HMG-CoA reductase), and inhibit the activity of acetyl-CoA carboxylase, malate enzyme, and ACAT. Harding et al. [43] fed hamsters with phytosterols (71% β-sitosterol, 15% β-sitostanol, 7% campestanol, and <1% stigmasterol); results showed that in the liver, the levels of SREBP 2 protein in the nucleus and the plasma cholesterol level both decreased significantly. Wistar rats and WKY rats fed with stigmasterol (0.5%) for 6 weeks showed that HMG-CoA reductase in liver was reduced significantly by 44% (Wister) and 77% (WKY) [44].

The cholesterol-lowering properties of phytosterols vary with their structure [54]. Structural differences in phytosterols lie in the chemical properties of the C ring, as well as the number and distribution of hydroxyl groups and other functional groups on the polycyclic rings, which significantly affect their absorption, metabolism, and bioactivities in vivo [55]. For example, the cholesterol-lowering effects of β-sitosterol and stigmasterol are stronger than those of campesterol [56]. Emerging evidence suggests that phytosterol esters exhibit an advantage over naturally occurring phytosterols in reducing atherosclerosis risk factors due to improving fat solubility and compatibility [57]. Consumption of phytosterol esters is more effective in reducing circulating cholesterol concentrations than that of free phytosterols [58].

### 3.2. Anticancer Effects

In recent years, the incidence of cancer is increasing, and research on cancer has attracted great attention. Recent epidemiological studies have found that the intake of phytosterols can reduce the incidence of common cancers such as breast cancer [59], gastric cancer [60], colon cancer [61], esophageal cancer [62], and ovarian cancer [63]. In a systematic meta-analysis, researchers aimed to systematically evaluate the results from previous studies focused on the understanding of phytosterols intake for cancer risk and found that there was an association between dietary phytosterols and cancer risk, but the linear association may be inconsistent for different phytosterols [64]. Similar results were seen in a recent meta-analysis, and the evidence provided by this study suggested that relatively modest (>200 mg) daily phytosterols impaired the development of tumors through hindering oncogenic signaling and suppressing multiple cancer hallmarks [65].

Presently, several possible anticancer mechanisms of phytosterols have been proposed, such as inhibiting the production of carcinogens; inhibiting the proliferation, invasion, and metastasis of cancer cells; inducing apoptosis and arresting the cell cycle (Table 2) [66]. In addition, angiogenesis is essential for cancer growth, and a previous study indicated that campesterol from Chrysanthemum coronarium L. has antiangiogenic activities. It was surmised that an antiangiogenic effect may be involved in the anticancer action of this phytosterol. Phytosterols could also regulate hormone levels in the body, thus affecting the growth of estrogen-dependent human breast cancer cells [67]. The mechanism of the tumor growth inhibitory effect of β-sitosterol may be related to protein kinase C (PKC) and the sphingomyelin cycle pathway. Phospholipase C is a key enzyme in the PKC pathway, and β-sitosterol can activate the sphingolipid cycle and may affect cell growth and apoptosis via ceramide [68]. Moreover, studies suggested that ceramide can activate the protein phosphatase 2A (PP2A), and β-sitosterol supplementation could improve in the PP2A activity [69].

### 3.3. Anti-Inflammatory Effects

Inflammation is the defense response to harmful stimuli of the immune system in the body. In recent years, more and more studies have indicated that many diseases such as cardiovascular disease, obesity, and arthritis are closely related to inflammation [79,80]. There have been various invitro studies and experimental animal models demonstrating that phytosterols have anti-inflammatory effects [81]. Over the past decades, phytosterols have been widely studied regarding their anti-inflammatory activity, and β-sitosterol, campesterol, and stigmasterol are the most commonly reported phytosterols found to have anti-inflammatory activities [82]. The signaling pathways involved in inflammation mainly include nuclear factor-additive B (NF-κB) and mitogen-activated protein kinase (MAPK) pathway. NF-κB is a class of inducible transcription factors that contribute to the production of inflammatory mediators; they normally bind to the inhibitor of NF-κB (IκB) as inactive complexes in the cytoplasm. After being stimulated, NF-κB members are activated and transferred to the nucleus for transcriptional function. The inflammatory responses are caused by the activation of NF-κB, through enhancing the IκB kinase β (IKKβ) function and the inhibitor of NF-κB 3 (IκB3) ubiquitination; as a result, NF-κB and TNF-ɑ increase, and TNF-ɑ has been confirmed to be involved in the pathogenesis of septic shock [83]. The MAPK pathway is an important signal pathway in eukaryotes that is evolutionarily conservative; the signaling pathways begin in the response of receptors and proteins on the cell membrane for stimulation in the external environment, and then the receptors and proteins interact with heterotrimer or monomer G protein coupled receptors, then activate MAPK [84]. MAPK, including p38, c-Jun amino-terminal kinase (JNK), and extracellular regulated protein kinase (ERK) MAPK, could transmit a variety of extracellular signals to intracellular, and participate in a variety of inflammatory cytokines expression. MAPK is regulated by the phosphorylation system, which phosphorylates specific serine and threonine substrates of their target proteins, and could activate the corresponding downstream protein expression, so as to control the release of various inflammatory factors. Moreover, the MAPK protein phosphorylation can lead to the activation of the NF-κB signaling pathway and the expression of nitric oxide synthase (iNOS) [85].

The anti-inflammatory mechanism of phytosterols may also be related to their ability to affect cell polarization. Kwanchanok Hunthayung et al. [86]. reported that cold-pressed rice bran oil noisome could reverse M1 proinflammatory macrophage transformations, and they hypothesized that phytosterols and other bioactive compounds in rice bran oil play a key role in this process. Jie et al. [87] confirmed that stigmasterol treatment attenuated M1 polarization in BV2 cells by reducing expression of M1 phenotype specific surface marker CD86. As nuclear receptors which could regulate the lipid metabolism, liver X receptors (LXRs) also acted as the regulators of anti-inflammatory responses. There were studies demonstrating that phytosterols were able to activate LXRs and relative pathways to exert anti-inflammatory effects [88,89]. However, it should be pointed out that not all experiments that evaluated the association between phytosterols and inflammatory variables (CRP, cytokines, and so on) could acquire positive results [90]. The anti-inflammatory effect of phytosterols appears to be influenced to a large extent by their structures. Yuan et al. [91] demonstrated that the position of the double bond and ethyl in phytosterol compounds may affect their anti-inflammatory activity; in addition, steric bulk, electronic, or hydrophobic features are also influencing factors (Figure 4).

### 3.4. Antioxidation Activities

The antioxidant activity of phytosterols has been confirmed and reported in numerous studies. As a natural free-radical scavenger, the peroxyl radical scavenging activity of β-sitosterol, first and foremost, is hydrogen transfer (HT) reaction when there is medium reactivity such as ·OOH and ·OOCH3. When the peroxyl radicals are significantly reactive, it can be explained by the radical adduct formation (RAF) mechanism [92]. Furthermore, phytosterols could not only increase the activity of antioxidant enzymes (SOD, CAT, GPx, etc.) but also reduce the level of peroxides (ROS, MDA, etc.) to alleviate oxidative stress [93,94]. It is worth mentioning that mitochondria are closely associated with oxidative stress, and in addition to being the main source of reactive oxygen species in cells, mitochondria are also one of the targets of reactive oxygen species attack [95]. In the mitochondrion, overaccumulation of ROS would cause the defects in DNA of mitochondria and respiratory complex to lead to impairing mitochondrial energy metabolism [96,97]. Zhang et al. demonstrated that phytosterols could have protective effects on mitochondrial damage by increasing the mitochondrial membrane potential and mitochondrial ATP content [98]. Sitosterol has been shown to induce mitochondrial uncoupling to reduce the extent of mitochondrial damage and upregulate glutathione redox cycling to protect against the oxidative stress [99].

### 3.5. Immunomodulatory Effects

Different studies demonstrated that with a shift towards T-helper (Th 1) cytokine profiles, phytosterols could enhance T-cell proliferation and natural killer cell activity, and phytosterols modulate the T-helper immune response in vivo [48]. β-Sitosterol is the most intensively studied phytosterol, and its immunomodulatory effects have been observed [100]. Fraile et al. [101] investigated the immunomodulatory properties of β-sitosterol in pig immune responses. The results showed that β-sitosterol activated swine dendritic cells and increased the viable peripheral blood mononuclear cell numbers. In addition, β-sitosterol enhanced the response of pigs to porcine reproductive and respiratory syndrome virus vaccine. Deeply, phytosterols influence the immunomodulatory activity by changing the presence of T-cells, which includes CD3+, CD4+, and CD8+. Deeper sight can be drawn to the expression of genes in biopsies [102]. Furthermore, phytosterols can regulate the sterol regulatory element binding proteins 1 and 2 (SREBP 1 and 2), which play an important part in the immunomodulatory activity related to CD8+ cells, and clonal amplification of virus will be significantly reduced by phytosterols during viral infection with SREBP signaling in CD8+ cells [103]. In addition, β-sitosterol could increase the phagocytic activity of pretreated differentiated U937 cells [104]. In another study, phytosterols was applied to aquaculture production, and phytosterols in C. cholla root can be used to enhance fish immunity [105].

### 3.6. Other Physiological Functions

Related studies found that phytosterols have some antibacterial activity in vitro [106,107]; for instance, phytosterols isolated from Laurencia papillosa red seaweed were confirmed to have antimicrobial activity against Gram-negative pathogenic bacteria [108]. Furthermore, there is various scientific evidence to demonstrate that phytosterols have a regulatory effect on gut microbes by modifying the composition of the gut microbes and the energy metabolism pathways [109,110]. In addition to the above activities, it was found that phytosterols have other aspects of physiological functions, such as antidiabetic [111], hormone-like effect [112], and blood lipid lowering effect [113]. The central nervous system (CNS) is the most cholesterol-rich organ in mammals; cholesterol homeostasis is essential for proper brain functioning, and dysregulation of cholesterol metabolism can lead to neurological problems [114]. Sharma et al. [115] reviewed β-Sitosterol, stigmasterol, and campesterol, and other minor sterols brassicasterol, lanosterol, 24(S)-saringosterol, and found that the 4,4-dimethyl sterols have regulatory effects on neurodegenerative diseases both in vivo and in vitro.

## 4. Application of Phytosterols

### 4.1. Application in Functional Foods

Phytosterols have many applications in food, for example, they can be used in order to develop nutraceuticals and functional foods with the potential to lower cholesterol levels, as natural antioxidants, and as fat substitutes [15]. Poulose et al. [116] extracted phytosterols from a seaweed and further nano-emulsified the stigmasterol and incorporated it into biscuits; the product was acceptable with good sensory and quality characteristics. Jaski et al. [117] extracted sunflower oil and olive leaf extract simultaneously to incorporate the active compounds in sunflower oil. Phytosterols were added to functional foods in free and esterified form. A special free-phytosterol premix was developed by Christiansen et al. [118], which can be dispersed into different foods. The physical state of this premix is microcrystalline, which comprises β-sitosterol, food-grade fats or oils, and water for the treatment of hypercholesterolemia. Recently, a light cream cheese spread of goat milk containing 6 g of phytosterols/100 g (LGC6) was developed by researchers [119]. LGC6 is not only a functional food with the potential to lower LDL cholesterol, but it also has more elastic and firm properties than phytosterol-free cheese spread, which can be used in the preparation of dishes or as a component of food products. In addition, Santos et al. [120] prepared nanostructured lipid carriers (NLCs) using crambe oils (CRs) to load free phytosterols. It was shown that the use of CRs allowed the development of NLCs with the most stable mixture of β forms and polymorphic forms β′ and β, which would make phytosterols available for better applications in foods requiring soft and creamy properties, such as margarine or spreads.

Due to the biological activities of phytosterol, application of phytosterol as a functional substance in food additive might be prospective [1,121]. An invention by Prakash and Dubois was related to a kind of functional sweetener, which consists of noncaloric or low-caloric potency sweeteners and phytosterols. It is believed that phytosterol-rich functional foods will become more popular as their functional properties become fully recognized.

Phytosterols have been widely used in the pharmaceutical industry due to their various physiological activities, including cholesterol-lowering, anticancer, and anti-inflammation effects [122]. Wang et al. [123] investigated the targeted intracellular anticancer drug, which was self-assembled phytosterol-alginate nanoparticles mediated by folate, and phytosterols exerted a positive effect on the whole anticancer drug system. In addition, a study of transdermal drug delivery system introduced the application of phytosterols, which were structured to improve the stimuli-responsive property of drugs in this system [124].

Phytosterols, especially β-sitosterol, have stable emulsification and strong surface activity; as a result, they can be used to replace cholesterol in skincare products in the cosmetic industry [125]. Furthermore, because of many biological activities in animals, phytosterols could be used as a new functional feed additive and have a broad application prospect in the feed industry.

### 4.2. Problems and Solutions in the Application of Phytosterols

#### 4.2.1. Phytosterol Oxidation Products (POPs)

Phytosterol-rich fat products such as vegetable oils and margarines are commonly used for cooking and baking; however, these heat treatments can promote the oxidation of phytosterols, resulting in the formation of POPs [126]. Furthermore, it has been suggested that POPs may be more atherogenic than cholesterol due to their molecular structure similarity [127]. A study on female LDL-receptor deficient mouse found that the POPs-containing diets enhanced atherosclerosis compared with control diet [128]. As a result, the investigation of POPs is ongoing. The increased phytosterol contents, heating temperature, and heating time could promote the formation of POPs in all tested fat-based products [127]. Compared with phytosterols in vegetable oils, phytosterols (naturally occurring or added) in margarine are not susceptible to oxidation. Under all experimental conditions applied, β-sitosterol is approximately 20% less sensitive to oxidation than campesterol. They finally concluded that the formation and composition of POPs was determined by the heating time, heating temperature, and initial contents of phytosterols [129,130].

Therefore, in fat-based products, the formation of POPs could be controlled through manipulating the heating time and temperature or changing the initial contents of phytosterols. In addition, it is reported that both heating time and surrounding medium can affect the extent of phytosterol degradation during the cooking process, which can impact the quality and safety of food product destined for cooking. Therefore, the impact of POPs on food safety and quality in food processing can be reduced through better control of processing methods, conditions, and surrounding environment [131].

#### 4.2.2. Bioaccessibility and Bioavailability

A boom has emerged in the addition of phytosterols to produce functional foods. Nevertheless, a major technical challenge has occurred—low bioaccessibility and bioavailability of phytosterols. Phytosterols are not soluble in water and are poorly soluble in fats and oils, which limits their application in food products [132]. As a result, improving their solubility is key to increasing the bioavailability and bioefficacy of phytosterols [133]. Different molecular structures corresponding to different physiochemical properties lead to the bioavailability of phytosterols [134] (Figure 5), including free sterols, sterol esters, steryl glycosides, and acylated glycosides [135]. In addition, the chemical and physical modification of plant sterols can help improve their bioavailability, The current research on chemical modifications is mainly focused on esterification to increase the oil solubility and water solubility for physical modifications (mainly microencapsulation) [9]. Additionally, chitosan, polymers, cyclodextrins, and dendrimers are commonly used to enhance the bioavailability of phytosterols [133,136,137]. For example, Ubeyitogullari et al. [137] proposed a novel approach to generate phytosterol nanoparticles with reduced crystallinity through the use of nanoporous starch aerogels, aiming to facilitate the addition of water-insoluble bioactives into low-fat foods to promote health and maximize the utilization of phytosterols. More approaches are being explored to improve the bioaccessibility and bioavailability of phytosterols and thus promote the widespread use of phytosterols.

## 5. Conclusions

With the development of phytosterol purification, structure analysis, synthesis, pharmacology, and clinical research, phytosterols have been widely applied in the food and pharmaceutical industries. Among them, phytosterols have attracted a lot of attention from scientists since the discovery of their biological activity in lowering plasma cholesterol levels in the last century. Nowadays, many bioactivities of phytosterols have been assessed in vivo and in vitro, while many of their pharmacological activities and mechanisms are still unclear, and in-depth investigation of the relationship between structure and function is also crucial. At the same time, new phytosterols with a variety of bioactivities will be isolated from different sources in the future. In addition, more attention should be paid to the chemical modifications of phytosterols. The application of phytosterols in the field of functional foods will be broadened based on the deeper understanding of their functional activities.

## Figures and Tables

**Figure 1 foods-13-01754-f001:**
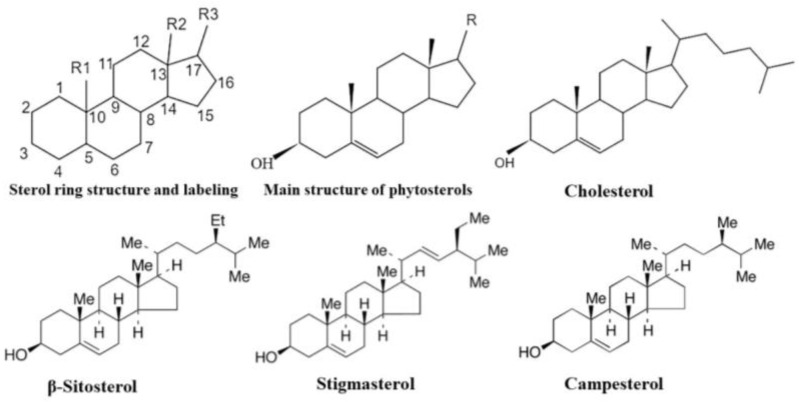
Basic skeleton of sterols and phytosterols, structure of cholesterol and major phytosterols (β-sitosterol, stigmasterol, and campesterol).

**Figure 2 foods-13-01754-f002:**
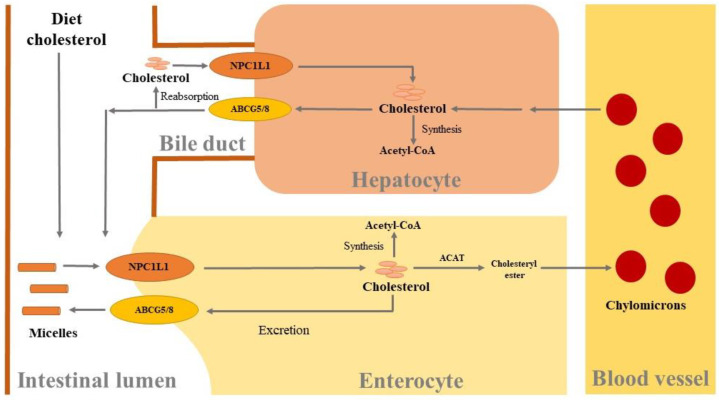
Absorption and metabolism of cholesterol in liver and intestines. Niemann-Pick C1-like 1 (NPC1L1) protein mediates cholesterol from the intestinal lumen into the hepatocytes and enterocytes. Most of the absorbed cholesterol is catalyzed and esterified by acyl-coenzyme A: cholesterol acyltransferase (ACAT) in the intestinal mucilage, leading to formation of cholesterol ester, and finally chylomicrons together with apolipoprotein, which enter the blood circulation through the lymphatic system. In addition, adenosine triphosphate binding cassette transporter G family 5/8 (ABCG 5/8) in hepatocytes and enterocytes is responsible for expelling small amounts of unabsorbed cholesterol back into the intestine lumen.

**Figure 3 foods-13-01754-f003:**
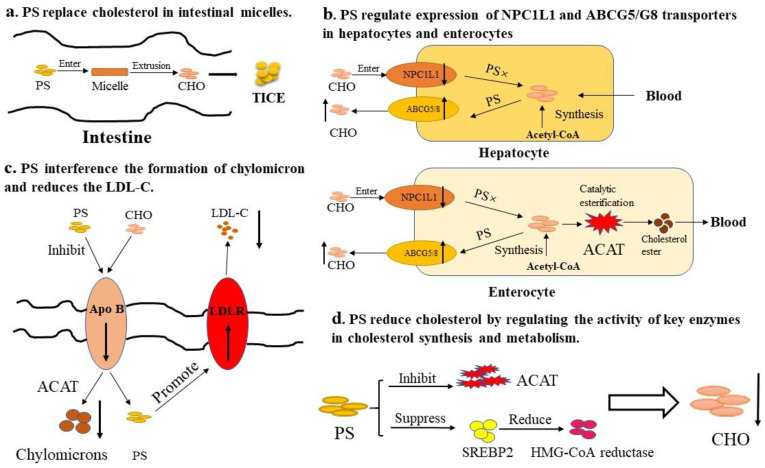
The mechanisms related to the cholesterol-lowering effects of phytosterols. (**a**) Phytosterols suppress the accumulation and absorption of cholesterol (CHO) by replacing it in the intestine micellar. (**b**) Phytosterols can regulate the expression of NPC1L1 and ABCG5/G8 transporters in hepatocytes and enterocytes. (**c**) Phytosterols interfere with the formation of chylomicron and increase the expression level of lipoprotein receptor (LDLR), thus removing LDL-C. (**d**) Phytosterols lower CHO by regulating the activity of key enzymes in CHO synthesis and metabolism. Phytosterols (PS); transintestinal cholesterol excretion (TICE); acyl-coenzyme A: cholesterol acyltransferase (ACAT).

**Figure 4 foods-13-01754-f004:**
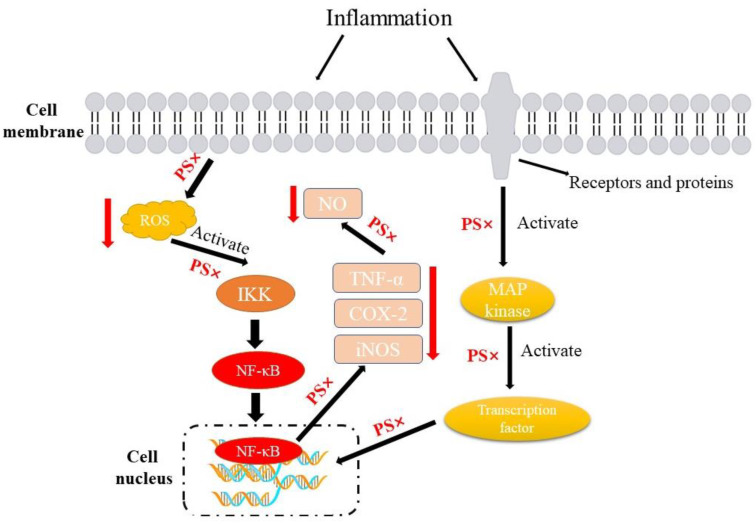
The mechanisms related to the anti-inflammatory activities of phytosterols. Firstly, phytosterols could inhibit the production of reactive oxygen species (ROS), thereby inhibiting the activation of IκB kinase (IKK) and further inhibiting the activity of NF-κB and its expression in the nucleus. At the same time, phytosterols could also inhibit the activation of MAP kinases via inflammatory stimulation through related receptors and proteins, thus preventing their regulation of related transcription factors. Finally, the production of inflammatory factors (TNF-α, COX-2, iNOS) is reduced and NO content is decreased. Phytosterols (PS).

**Figure 5 foods-13-01754-f005:**
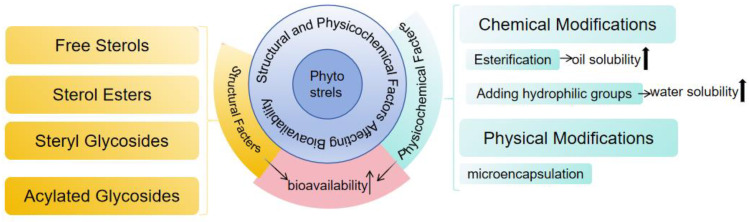
The structural and physicochemical factors affecting the bioavailability of phytosterols.

**Table 1 foods-13-01754-t001:** The cholesterol-lowering effects of phytosterols and their related mechanisms.

Phytosterols	Dose	Model	Mechanisms	References
Sitosterol	250 μmol/L	FHs 74 Int cells	Reduces the mRNA levels of NPC1L1 and HMG-CoA reductase.	[45]
Sitosterol and stigmasterol	100 μmol/L	Caco-2 and HepG2 cells	Reduces the messenger RNA expression levels of NPC1L1, scavenger receptor class B type I, LDLR, and HMG-CoA reductase.	[46]
Stigmasterol, campesterol and β-sitosterol	100 μmol/L	Caco-2 cells	Decreases apolipoprotein secretion of ApoB48.	[47]
Sitosterol, sitostanol and fucosterol	125, 250 μmol/L	Caco-2 cells	Activates LXR and its target gene ABCA1 to increase reverse transport.	[37]
Schottenol and spinasterol	2.5, 10 μmol/L	BV cells	Activates LXRα, LXRβ, and target genes ABCA1, ABCG1.	[48]
Saringosterol	30 μmol/L	HEK293T, HepG 2, THP-1, RAW264.7 and Caco-2 cells	Activates the target gene of LXR (especially LXRβ).	[49]
Phytosterol mixture ^a^	2% of feed	Female ApoE-/- mice	Reduces cholesterol absorption in enterocytes.	[50]
β-Sitosterol and stigmasterol	0.1% of feed	Male hamsters	Decreases mRNA levels of ACAT and MTP in intestine and reduces chylomicron assembly.	[51]
Phytosterol mixture ^b^	20 g/kg	Male hamsters	Inhibits the expression of SREBP-2 and increases the ABCG5 level in liver.	[43]
Stigmasterol	0.5% of feed	Wistar and WKY rats	Inhibits activity of HMG-CoA reductase.	[44]
Phytosterol ester mixture ^c^	50 mg/every mouse	C57BL/6J mice	Reduces expression of SREBP2 and HMG-CoA reductase.	[52]
Phytosterol mixture ^d^	1%, 2%, 4% or 8% of feed	Wild-type and Abcg5-/- mice	Improves expression of ABCG5 to increase excretion of TICE.	[39]
Phytosterol mixture ^e^	2.13 g/kg	Male SHRSP and WKY inbred rats	Increases ABCG8 level of intestine, mRNA of ABCA1 and ABCG5 in liver, and increases the mRNA expression of 27α hydroxylase (CYP27A1).	[53]

^a^ 41% β-sitosterol, 20% campesterol, and 22% stigmasterol of feed; ^b^ 71% β-sitosterol, 15% β-sitostanol, 7% campestanol, and <1% stigmasterol; ^c^ 70% sitostanol ester, 30% campestanol ester; ^d^ 69% β-sitosterol, 15.7% campesterol, and 15.7% sitostanol; ^e^ 22.0% brassicasterol, 31.9% campesterol, and 43.2% β-sitosterol.

**Table 2 foods-13-01754-t002:** The anticancer effects of phytosterols and their related mechanisms.

Phytosterols	Dose	Types of Cancer	Mechanisms Related to Anticancer Effects	References
Phytosterols ^a^	115, 11, 6 μM	Colon cancer	Influence on cell viability and cell cycle.	[70]
Phytosterols ^b^	2% of feed	Breast cancer	Induction of lipoprotein oxidation.	[71]
Phytosterols ^c^	16 μM	Prostate cancer	Inhibition of the expression of caveolin-1.	[72]
Phytosterols ^d^	2.80–467.11 μg/mL	Leukemia	Inhibition of leukemic cell proliferation.	[4]
Phytosterols ^e^	2.5–25 μg/mL	Breast cancer	Suppression of tumor growth and the expression of tumor markers.	[64]
Phytosterols ^f^	0.1–2% of feed	Colon cancer	Slows tumor proliferation.	[70]
Phytosterols	13.2 μM	Colon cancer	Increases the number of cells in sub-G1 phase.	[73]
β-Sitosterol	13, 26, 52 μM	Breast, colon and cervical cancer	Induction of DNA fragmentation and apoptosis.	[74]
β-Sitosterol	2.5–25 μM	Colon cancer	Induces apoptosis by increasing the sub-G1 cell population.	[75]
Campesterol	30, 60, 120 μM	Lymphoma cancer	Enhancing cell apoptosis.	[76]
Ergosterol	20 μM	Human lung adenocarcinoma cells	Inhibition of cancer growth (the oxidation products of ergosterol).	[77]
β-Sitosterol, stigmasterol	>56.0 mg/day or >9 mg/day	Gastric and stomach cancer	Affecting testosterone metabolism and enhancing apoptosis of cancer cells.	[78]

^a^ Phytosterols: 115 μM β-sitosterol + 11 μM campesterol + 6 μM stigmasterol = 132 μM; ^b^ Phytosterols were composed of 20% campesterol, 22% stigmasterol, 41% β-sitosterol, and others; ^c^ Phytosterols were composed of 10% campesterol, 75% β-sitosterol, and others; ^d^ Phytosterols were extracted from black rice bran and mainly composed of β-sitosterol, stigmasterol, and campesterol; ^e^ Phytosterols were extracted from sweet potato; ^f^ Phytosterols: 60% β-sitosterol, 30% campesterol, and 5% stigmasterol.

## Data Availability

The original contributions presented in the study are included in the article, further inquiries can be directed to the corresponding author.

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
