# Peer review of "Phytosterols: Physiological Functions and Potential Application"

_foods, 2024, doi:10.3390/foods13111754_

Round 1

Reviewer 1 Report

Comments and Suggestions for Authors

This narrative review [29% ≥10 y old (41/139)] addresses three important topics related to phytosterols: A) Chemical/analytical aspects (Sections 1,2), B) Health promoting-effects (Section 3: Hypocholesteremia, anti-cancer, anti-inflammatory, antioxidant, immunomodulatory, other) and C) Application (Section 4: Use in functional foods, oxidation problems, gastrointestinal fate). The weight of each of these sections of the global text is B>C>A and supported with two systematic tables (Hipocholesterolemic and anti-cancer effects) and 4 figures (1. Chemical structures, 2. First-pass metabolism, 3. Cholesterol-reducing mechanisms, 4. Anti-inflammatory effect).

Although the presented information seems to be adequate and relatively new, quite similar yet more comprehensive reviews on this subject (doi: 10.1002/ptr.7312) have been recently reported (some of which were not included in this new review article  (doi: 10.1080/10408398.2021.1888692). Nevertheless, the manuscript´s uniqueness and scientific soundness could be improved if the following adjustments are considered:

General. A) The use of abbreviations should be reduced as much as possible, B) English grammar/syntax should be reviewed by a native English speaker or by a formal translation agency. B) Italics should be used when needed (e.g. in vivo, in vitro)

Abstract. A) Could be more specific to relevant findings and novel scientific fields (to support the manuscript’s uniqueness/contribution).

Introduction. OK

Body of text. A) Section 1 should also include two subsections: Sources and analytical identification, B) Section 2 should emphasize emerging health-promoting bioactivities that were either scarcely discussed (e.g. phytosterols and immunomodulation) or not included (e.g. neurodegenerative disorders, C) Considering that most if not all ultimate bioactivities depend on the gastrointestinal/metabolic fate of dietary phytosterols, authors should back up section 4.2.2. with a figure or a brief systematic review on pharmaceutical preparations to enhance phytosterol bioaccessibility and bioavailability.

Figures. A) The sharpness and size of all figures should be improved in accordance with the journal's guidelines, B) A new figure summarizing the structural and physicochemical factors affecting the bioaccesibility and bioavailability of dietary phytosterols, is strongly suggested.

References. A) That seems like a lot of references even for a systematic review article. Suggestion: Reduce to 100 (75% of ≤10y). B) Authors should take a second look on the format of specific references (e.g. 91, 93, 98, 126), C) Reduce self-citations to say 3 references.

Comments on the Quality of English Language

Moderate changes are needed

Author Response

Comments 1: The use of abbreviations should be reduced as much as possible.

Response 1: Thank you for pointing this out. The modification has been made in accordance with the requirements, we try to avoid the use of abbreviations in the manuscript and provide corresponding explanations for all abbreviations involved.

Comments 2: English grammar/syntax should be reviewed by a native English speaker or by a formal translation agency.

Response 2: Thank you for taking your time to review our manuscript. According to your valuable comments, the manuscript have been carefully revised. The revision were given directly in red font in revised manuscript.

Comments 3: Italics should be used when needed (e.g. in vivo, in vitro).

Response 3: Thanks a lot. We've checked the full text and italicised what should be italicised. Please see lines 175, page 5, line 228, page 7, line 301, page 9, line 319, page 10, line 424, page 12.

Comments 4: Abstract could be more specific to relevant findings and novel scientific fields (to support the manuscript’s uniqueness/contribution).

Response 4: Thank you for taking your time to review our manuscript. According to your valuable comments, the Abstract have been carefully revised. Please see lines 7-19 ,page 1 .

Comments 5: Introduction. OK

Response 5: Thank you very much for your approval of the manuscript.

Comments 6: Section 1 should also include two subsections: Sources and analytical identification.

Response 6: Thank you for pointing this out. The modification has been made in accordance with the requirements. Please see lines 58-80, page 2.

Comments 7: Section 2 should emphasize emerging health-promoting bioactivities that were either scarcely discussed (e.g. phytosterols and immunomodulation) or not included (e.g. neurodegenerative disorders.

Response 7: Thank you for pointing this out. The modification has been made in accordance with the requirements. Please see lines 327-332, page 10.

Comments 8: Considering that most if not all ultimate bioactivities depend on the gastrointestinal/metabolic fate of dietary phytosterols, authors should back up section 4.2.2. with a figure or a brief systematic review on pharmaceutical preparations to enhance phytosterol bioaccessibility and bioavailability.

Response 8: Thank you for pointing this out. The modification has been made in accordance with the requirements. Please see lines 404-417, page 12.

Comments 9: The sharpness and size of all figures should be improved in accordance with the journal's guidelines.

Response 9: Thank you for pointing this out. The sharpness and size of all figures have been improved in accordance with the journal's guidelines.

Comments 10: A new figure summarizing the structural and physicochemical factors affecting the bioaccesibility and bioavailability of dietary phytosterols, is strongly suggested.

Response 10: Thank you for your suggestion. We have added the structural and physicochemical factors affecting the bioaccesibility and bioavailability of phytosterols in the manuscript. Please see lines 404-417, page 12.

Comments 11: That seems like a lot of references even for a systematic review article. Suggestion: Reduce to 100 (75% of ≤10y).

Response 11: Thank you for pointing this out. The number of references has been reduced as much as possible, which may still exceed 100. However, we can guarantee that the number of references in the past decade is greater than 75%.

Comments 12: Authors should take a second look on the format of specific references (e.g. 91, 93, 98, 126).

Response 12: Thank you for pointing this out. According to your valuable comments, the format of references have been carefully revised.

Comments 13: Reduce self-citations to say 3 references.

Response 13: Thank you for pointing this out, We have reduced the self-citations to 2 references.

Reviewer 2 Report

Comments and Suggestions for Authors

This review descrives the potential uses of phytosterols, deals with how they are extracted, the several health benefits they provide and how they can be use on food to increase the beneficial effects.

Globally is well structured and with quite good references, except in the beginning of the introduction, presenting of the phytosterols, also when it comes to the extraction of sterols there is a limited number of references as well as when talking about food enhanced with phytosterols. The references and the writting is complete and clear on the "health" aspects.

But should be completly review in the extraction section, where it lacks of references and broader perspective.

In the "food" section, it's certainly difficult to find out scientific litterature talking about the subject but probably there are more than those 2-3 cited, which is too reduce.

There are quite a lot references which are not adecuated to the statement associated, most probably only talk about the topic, but it's not the major research objectives, being then not appropriated to be cited. Sometimes, the reference only talks about a small part of the topic, being the sentence too large to be exclusively covered by just this reference.

In the attached document you find the detailed references and parts to be corrected.

To sum up, the structure is fine, references should be checked and changed to more fair and appropriated. The sections about extraction and food should be completed and reinforced with more litterature.

Abstract is ok

Conclusions are ok

Comments on the Quality of English Language

Author Response

Comments 1: This review describes the potential uses of phytosterols, deals with how they are extracted, the several health benefits they provide and how they can be use on food to increase the beneficial effects.

Response 1: Thank you for taking your time to review our manuscript. According to your valuable comments, the manuscript have been carefully revised. The revision were given directly in red font in revised manuscript.

Comments 2: Globally is well structured and with quite good references, except in the beginning of the introduction, presenting of the phytosterols, also when it comes to the extraction of sterols there is a limited number of references as well as when talking about food enhanced with phytosterols.

Response 2: Thanks a lot. According to your valuable comments, the manuscript have been carefully revised. Please see lines 58-80, page2.

Comments 3: The references and the writting is complete and clear on the "health" aspects. But should be completly review in the extraction section, where it lacks of references and broader perspective.

Response 3: Thank you for your valuable comments, the extraction section has been carefully revised. Please see lines 58-80, page2.

Comments 4: In the "food" section, it's certainly difficult to find out scientific literature talking about the subject but probably there are more than those 2-3 cited, which is too reduce.

Response 4: Thanks a lot. the "food" section has been carefully revised. Please see lines 335-341, page 10.

Comments 5: There are quite a lot references which are not adecuated to the statement associated, most probably only talk about the topic, but it's not the major research objectives, being then not appropriated to be cited. Sometimes, the reference only talks about a small part of the topic, being the sentence too large to be exclusively covered by just this reference.

Response 5: Thank you for your constructive proposal. We have carefully checked the full text and amended the references as required to make them more accurate.

Comments 6: In the attached document you find the detailed references and parts to be corrected.

Response 6: Thank you very much for your approval of the manuscript. It has been modified according to your request.

Comments 7: To sum up, the structure is fine, references should be checked and changed to more fair and appropriated. The sections about extraction and food should be completed and reinforced with more literature.

Response 7: Thank you for your constructive proposal. We have carefully checked the full text and amended the references as required to make them more accurate. The revision were given directly in red font in revised manuscript.

Comments 8: Abstract is ok

Response 8: Thank you very much for your approval of the manuscript. 

Comments 9: Conclusions are ok

Response 9: Thank you very much for your approval of the manuscript.

Round 2

Reviewer 1 Report

Comments and Suggestions for Authors

A) The changes in the new version of the manuscript (highlighted in red) could not be traced with those indicated in your response letter. 

B) Several changes were not sufficiently detailed. For example, the suggestion to make "subsections" (naturals sources and analyticial identification) in section 2 were just adressed with two small paragraphs. By the way, why did section 2 decreased instead of increased  as recommended?.

C) The suggested figures were not included (for example, the graphical summary was not included and they simply made changes to the text that are not enough to understand their statements) 

C) The English syntax and grammar did not improve. You must send the manuscript to a formal translation agency and provide it with the correponding certificate, 

D) It is strongly sdviced to prepare a more careful point-by-point letter and give specific justifications for unaddressed requests.

Comments on the Quality of English Language

English syntax and grammar must be reviewed once again 

Author Response

Comments 1: The changes in the new version of the manuscript (highlighted in red) could not be traced with those indicated in your response letter.

Response 1: Thank you for pointing this out. We are very sorry for this. But I don't know why the serial number changed after uploading, I will attach the modified parts to the response.

Comments 2: Several changes were not sufficiently detailed. For example, the suggestion to make "subsections" (naturals sources and analyticial identification) in section 2 were just addressed with two small paragraphs. By the way, why did section 2 decreased instead of increased  as recommended?.

Response 2: the section of Extraction of Phytosterols has been addressed with two small paragraphs. Please refer to lines 54-96, pages 2-3..

Extraction of Phytosterols

2.1. Sources of Phytosterols and Extraction Techniques

2.2. Analytical Identification of Phytosterols

Comments 3: The suggested figures were not included (for example, the graphical summary was not included and they simply made changes to the text that are not enough to understand their statements) .

Response 3: Thanks a lot. A new figure summarizing the structural and physicochemical factors affecting the bioaccesibility and bioavailability of dietary phytosterols has been added in 4.2.2. Please see Fig. 5, lines 409-430, page 12.

Figure 5. The structural and physicochemical factors affecting the bioavailability of phytosterols

Comments 4: The English syntax and grammar did not improve. You must send the manuscript to a formal translation agency and provide it with the correponding certificate.

Response 4: Thank you for taking your time to review our manuscript. According to your valuable comments, the English syntax and grammar have been carefully revised.

Comments 5: It is strongly adviced to prepare a more careful point-by-point letter and give specific justifications for unaddressed requests.

Response 5: Thank you for your suggestion. We have made the modifications according to your suggestions.

Reviewer 2 Report

Comments and Suggestions for Authors

The authors have now provided a corrected version of the manuscript, they have done a good job, adding content and references to the most incomplete ones.

There are still few references which need to corrected, mainly the first 4 and the 13th. For the first 4 it was already requested in my previous review.

Please see attached the comments

Author Response

Comments 1: The authors have now provided a corrected version of the manuscript, they have done a good job, adding content and references to the most incomplete ones.

Response 1: Thank you very much for your approval of the manuscript. According to your valuable comments, the manuscript has been carefully revised. The revision was given directly in red font in revised manuscript.

Comments 2: There are still few references which need to corrected, mainly the first 4 and the 13th. For the first 4 it was already requested in my previous review. Please see attached the comments.

Response 2: Thanks a lot. According to your valuable comments, the manuscript have been carefully revised. Please see lines 23-29, page 1.